# Pathological Changes and Metabolic Adaptation in the Myocardium of Rats in Response to Chronic Variable Mild Stress

**DOI:** 10.3390/ijms25115899

**Published:** 2024-05-28

**Authors:** Marta Ostrowska-Leśko, Mariola Herbet, Kamil Pawłowski, Agnieszka Korga-Plewko, Ewa Poleszak, Jarosław Dudka

**Affiliations:** 1Department of Toxicology, Medical University of Lublin, 8b Jaczewski Street, 20-090 Lublin, Poland; mariola.herbet@umlub.pl (M.H.); jaroslaw.dudka@umlub.pl (J.D.); 2Independent Medical Biology Unit, Medical University of Lublin, 8b Jaczewski Street, 20-090 Lublin, Poland; 3Department of Applied Pharmacy, Medical University of Lublin, 1 Chodźko Street, 20-093 Lublin, Poland

**Keywords:** chronic variable mild stress, oxidative stress, metabolism, adverse effects

## Abstract

Chronic variable mild stress (CVS) in rats is a well-established paradigm for inducing depressive-like behaviors and has been utilized extensively to explore potential therapeutic interventions for depression. While the behavioral and neurobiological effects of CVS have been extensively studied, its impact on myocardial function remains largely unexplored. To induce the CVS model, rats were exposed to various stressors over 40 days. Behavioral assessments confirmed depressive-like behavior. Biochemical analyses revealed alterations in myocardial metabolism, including changes in NAD+ and NADP+, and NADPH concentrations. Free amino acid analysis indicated disturbances in myocardial amino acid metabolism. Evaluation of oxidative DNA damage demonstrated an increased number of abasic sites in the DNA of rats exposed to CVS. Molecular analysis showed significant changes in gene expression associated with glucose metabolism, oxidative stress, and cardiac remodeling pathways. Histological staining revealed minor morphological changes in the myocardium of CVS-exposed rats, including increased acidophilicity of cells, collagen deposition surrounding blood vessels, and glycogen accumulation. This study provides novel insights into the impact of chronic stress on myocardial function and metabolism, highlighting potential mechanisms linking depression and cardiovascular diseases. Understanding these mechanisms may aid in the development of targeted therapeutic strategies to mitigate the adverse cardiovascular effects of depression.

## 1. Introduction

Chronic variable mild stress (CVS) is a well-known rat paradigm model. The primary purpose of using such a model is to induce basic behavioral impairments manifested as depressive-like behavior to improve screening tests for possible therapeutic regimens for depression treatment [1,2]. Willner improved the method after Katz first proposed it, providing comprehensive behavioral and neurobiological outcomes concerning depressive symptomatology [2,3]. Briefly, adolescent animals are chronically exposed to various micro-stressors unpredictably, resulting in reduced rat locomotor activity, change in daily rhythm, relative weight loss, and altered cytokine levels. These changes are reversible after antidepressant treatment.

Cardiovascular diseases are the leading cause of death worldwide, with approximately 17.9 million deaths per year, which is approximately 32% of all global deaths [4]. Among these fatalities, 85% are attributed to heart attacks and strokes. Chronic stress is a significant risk factor for cardiovascular disease. Research indicates that elevated levels of stress significantly raise the risk of heart disease by 27% [5]. People who have depression have a 64% higher chance of developing coronary artery disease [6]. Numerous studies indicate a significant role of psychosocial factors in the etiology and progression of cardiovascular diseases. Indeed, the COVID-19 pandemic and its associated social distancing regimen increased the incidence of major depressive disorder [7,8]. Compelling evidence links depressive disorders with the increased probability of developing coronary heart disease and acute cardiovascular consequences, including myocardial infarction [9,10,11,12], and there are findings supporting the belief that long-term stress is linked to abnormalities in the autonomic nervous system of the heart [13,14,15]. The mechanisms underlying this relationship are not yet fully understood; moreover, most studies on mechanisms of cardiotoxicity in depressed patients are documented based on epidemiological observational study designs [16,17,18]. The relationship between depression and social isolation—which worsens prognosis and quality of life and increases mortality in patients with cardiovascular disease—has also been proven [18]. Another indirect piece of evidence of the relationship between depression and heart disease may be the reduction in risk of occurrence when depressed patients are treated with antidepressants [19]. It is essential not only to treat depression but also to pay attention to concomitant diseases and to take protective actions; therefore, recognizing and understanding the molecular mechanisms underlying this relationship is essential to determine the direction of these changes. This is crucial in understanding how chronic stress affects heart changes.

To address these fundamental questions, we have exposed rats to CVS procedures, which is a widely accepted method of inducing changes that reflect depression-like behavior in rodents. This study is the first to thoroughly evaluate the connections between different components in the myocardium at the molecular and biochemical levels, despite numerous previous research on the CVS model. This work comprehensively examines genes related to glucose metabolism, lipid oxidation, oxidative stress, cardiac remodeling, amino acids, and biochemical and histological investigation. In the future, understanding the interrelationship of these elements may help explain causes, prevent heart illnesses, and develop novel therapeutic strategies. Results from the conducted study may improve understanding of how depression and prolonged chronic stress increase the risk of cardiovascular disease. Currently, no successful strategies prevent cardiac remodeling caused by chronic stress.

## 2. Results

### 2.1. Establishment of the CVS Model

First of all, we performed a forced swim test (FST) to confirm that the 40-day CVS procedure induced anhedonia and depressive-like behavior in rats. As shown in Figure 1a, the immobility time was prolonged in rats subjected to CVS procedures (CMS) in comparison to the control (CTL) group [*p* ≤ 0.01; t = 4.951 (18)], which confirmed the efficiency of model induction.

Moreover, a significant (almost two-fold) increase in the median blood fasting glucose concentration in rats exposed to CVS was demonstrated (U = 0; *p* ≤ 0.001; Figure 1b); this is a well-known cortisol-mediated response to urgent stress situations. Taking together, a depressive-like behavior was induced, which confirms the successful establishment of the CVS model.

### 2.2. Body Weight Assessment

A Student’s *t*-test showed significantly lower mean body weight in the CMS group on the 20th [*p* ≤ 0.001; t = 10.83 (18)] and 40th [*p* ≤ 0.001; t = 8.788 (18)] day as compared to the CTL group (Figure 2a). The heart/total body weight ratio on the 40th day of the experiment was significantly higher (*p* ≤ 0.001; t = 7.363 (12)] (Figure 2b).

### 2.3. Biochemical Analysis

In this experiment, we evaluated NAD^+^, NADH, NADP^+^, and NADPH concentrations, and Ldha, Ldhb, and NT-proBNP protein content in rats’ hearts homogenates as well as fasting glucose (data provided in Section 2.1) and lactate concentrations in blood serum. A Mann–Whitney U test showed that the NAD^+^ concentration was significantly lower in the hearts of rats subjected to the CVS procedure (U = 34.50; *p* ≤ 0.01; Figure 3a), whereas NADH concentration did not change (U = 107.5; *p* ≥ 0.05; Figure 3b). In turn, we demonstrated a significantly lower concentration of NADP^+^ [*p* ≤ 0.01; t = 3.032 (25); Figure 3d] and an increased concentration of total NADPH (U = 8; *p* ≤ 0.001; Figure 3e) and NADPH/NADP ratio (U = 4; *p* ≤ 0.001; Figure 3f) in the CMS group. No significant difference between the study groups was observed in lactate concentration (U = 46; *p* ≥ 0.05; Figure 3c).

The Mann–Whitney U test showed that NT-proBNP significantly elevated after the CVS procedure (U = 0; *p* ≤ 0.001; Figure 4a). Ldha and Ldhb content analysis showed no significant differences between the study groups [Ldha, U = 12; *p* ≥ 0.05; Figure 4b; Ldhb, *p* ≥ 0.05; t = 0.4554 (18); Figure 4c].

### 2.4. Free Amino Acid Analysis

The analysis of free amino acid concentrations in the heart homogenates of rats subjected to the CVS procedures showed a statistically significant increase in the following amino acids: asparagine (U = 2; *p* ≤ 0.05), valine (U = 2; *p* ≤ 0.01), methionine (U = 0; *p* ≤ 0.01), and proline (U = 24; *p* ≤ 0.01) (Table 1). In turn, significantly reduced levels of phosphoserine (U = 3; *p* ≤ 0.05), arginine (U = 5; *p* ≤ 0.05), and ethanolamine (U = 2; *p* ≤ 0.01) were observed.

### 2.5. Oxidative DNA Damage

The oxidative DNA damage was evaluated by measuring the number of abasic (AP) sites. AP sites are one of the major types of DNA damage generated by reactive oxygen species (ROS). A Student’s *t*-test showed a significantly increased number of AP sites [*p* ≤ 0.01; t = 4.521 (8)] in the DNA of rats subjected to CVS procedures in comparison with the control group (Figure 5).

### 2.6. Molecular Analysis

The analysis of mRNA gene expression level showed significant changes in the pathways associated with glucose transport and metabolism, gluconeogenesis, glutaminolysis, pyruvate metabolism, monocarboxylate transport, urea cycle, lipid transport, synthesis and oxidation, oxidative stress, and heart remodeling (Table 2). The most overexpressed genes were: *Hk2* (U = 50; *p* < 0.000), *Fbp2* (U = 101; *p* < 0.000), *G6pc* (U = 60; *p* < 0.000), *Slc1a5* (U = 40; *p* < 0.000), *Pdk4* (U = 4; *p* < 0.000), *Slc16a1* (U = 22; *p* < 0.000), *Prkaa2* (U = 16; *p* < 0.000), *Cpt1a* (U = 5; *p* < 0.000), *Ppargc1a* (U = 4; *p* < 0.000), *Myh6* (U = 21; *p* < 0.000), and *Myc* (U = 0; *p* < 0.000). *Fbp1* (U = 1; *p* < 0.000) and *Myh7* (U = 74; *p* < 0.000) were down-regulated.

### 2.7. Histological Staining

Table 3 demonstrates the presence and intensity of minor and moderate morphological changes after the CVS procedure; we did not observe major intensity changes.

H + E staining showed increased acidophilicity of the cells, wavy course of fibers, and infiltration of the mononuclear cells (Figure 6).

Van Gieson’s staining revealed an increase in collagen deposition surrounding the intracoronary blood vessels of rats subjected to the CVS procedures (Figure 7). 

The collagen extended into the myocardial interstitium. In turn, PAS and PAS with diastase staining showed glycogen deposition in rats with induced depressive-like behavior (Figure 8).

## 3. Discussion

Our study is the first to comprehensively evaluate the cardiotoxicity of chronic variable stress in a rat model of depression at the biochemical and molecular levels. Understanding the interdependence of these factors may explain the causes of emerging disorders and may contribute to the development of new preventive and therapeutic approaches to the co-occurrence of depression and cardiovascular diseases. Currently, there are no effective strategies to avoid the changes in the structure and function of the heart induced by long-term stress.

Despite the well-documented role of cardiovascular disease in the pathogenesis of depression [20,21,22], the reversible mechanisms of cardiotoxicity development in response to chronic stress are poorly understood. Most often, the etiology of changes in the myocardium has been based on assumptions resulting from indirect evidence—the action of the stress hormone (cortisol), especially in Cushing’s syndrome [23,24], and pharmacological activation of the sympathetic system [25,26]. Chronic hypoxia, disruptions in the utilization of glucose, as well as the presence of oxidative stress could result in cardiac energy deficit and predispose to cardiac dysfunction. Under unfavorable conditions, the heart adapts to maintain cardiac metabolic efficiency and ATP production. The flexibility in the switch of metabolic fuels underlies cellular adaption. This research confirmed the direct influence of chronic stress on changes in the myocardium and allowed for a deeper recognition of their nature. Below, we provide direct evidence for a relationship between chronic stress and the development of pathological myocardial lesions assessed at the molecular, biochemical, and tissue levels and discuss induced changes in the hearts of rats with depressive-like behavior.

Changes in the proportion of aerobic and anaerobic pathways in energy generation are associated with the pathogenesis of cardiac diseases [27,28]. However, it is essential to note that an elevation in glycolysis activity, which is the anaerobic metabolic pathway, can occur without being solely attributed to hypoxia [27,28]. We assessed how the CVS affects heart muscle glycolysis activity. Despite the *Ldha* gene overexpression, Ldh-A protein content, which converts pyruvate to lactate, did not alter. It is important to take into account that the LDHA subunit serves as both a constituent of the LDH enzyme and a transcription factor for other enzymes involved in the glycolytic process. We also observed no changes in the Ldh-B protein catalyzing the reverse conversion. No serum lactate rise—a sensitive indicator of hypoxia—was observed. Overexpression of *Pdk4* demonstrated in this study may increase fatty acid (FA) oxidation and a reduction in glycolysis [29]. PDK4 regulates the activity of pyruvate dehydrogenase (PDH) by inhibiting it. This inhibition increases acetyl-CoA flow from β-oxidation into the TCA cycle, resulting in enhanced FA oxidation and decreased glycolysis or the redirection of glycolytic intermediates to other metabolic pathways [30].

Glycolysis is an oxygen-independent ATP source. Analysis showed higher glycolysis-related expression of genes, suggesting pathway activation. Heart dysfunction is characterized by mitochondrial malfunction and a shift from using FAs to glucose for energy production. The heart’s response to chronic stress involves changes in amino acid metabolism, especially glucogenic amino acids that can be converted into intermediates of the citric acid cycle, leading to the production of glucose through gluconeogenesis. Moreover, certain amino acids may accumulate in heart failure, ischemic heart disease, or cardiomyopathies [31]. Here, we measured free amino acid concentrations to provide insights into the heart’s response. In the chronic unpredictable mild stress (CUMS) model, the amino acids were reported to be altered in the heart tissue [32]. While the changes in concentration levels of amino acids observed in this study were not discussed concerning the heart, some general observations can be made. An increase in concentration of almost all glucogenic amino acids indicates a shift in metabolism towards producing glucose, either to meet energy demands or pathological factors as a response to the CVS. Moreover, we observed overexpression of *Prkaa2*, *G6pc*, and *Fbp2* genes, indicating gluconeogenesis pathway activation.

Overexpression of *Pdk4* may reduce pyruvate-to-acetyl-coenzyme A conversion. If NAD+ is available, glycolysis occurs. We found reduced NAD+ levels, which may indicate greater cardiomyocyte ATP demand. Moreover, *Pdk4* plays a crucial role in regulating metabolism by facilitating the shift from glucose to FAs as the primary energy source. Protein Kinase AMP-Activated Catalytic Subunit Alpha 2 (AMPKα2) (which encodes *Prkaa2* gene) activity is influenced by the cellular energy state and it restores energy balance after shortages [33]. AMPKα2 is crucial for glycogen production. Consistent with these observations are PAS and PAS with diastase stainings, whose evaluation revealed glycogen deposition in the myocardium (Figure 8); this may be additionally linked to high serum glucose. These findings may suggest glycolysis redirects to glycogen production and activates the pentose–phosphate (PP) pathway. The increase in NADPH and decrease in the NADPH/NADP+ ratio support these hypotheses.

Then, we assessed whether the CVS procedure causes oxidative stress in the myocardium. An imbalance between the intensity of oxidative processes that stimulate the generation of ROS and antioxidant mechanisms leads to oxidative stress [34,35,36]. Excess ROS damage DNA, including oxidative base damage and abasic (AP) sites [34,37]. This study indicated a considerable rise in DNA AP sites, which indicate oxidative DNA damage by chronic stress factors. Inflammation, fibrosis, and necrosis can therefore result from the resulting oxidative damage [38,39,40]. Necrosis and inflammation were present in H + E staining (Figure 6). One of the consequences of oxidative stress in cells is the formation/intensification of inflammatory changes. Many studies have revealed a connection between oxidative–antioxidant imbalance and immune processes [36]. It can therefore be assumed that the observed changes may be indirectly related to oxidative stress in rat heart muscle cells exposed to chronic stress; however, further research is required to precisely explain the mechanisms behind the changes. In rats subjected to the CVS procedure, van Gieson’s staining showed increased collagen deposition around intracoronary blood vessels, which extended into the cardiac interstitium (Figure 7). Increased collagen deposition reduces flexibility and signals arterial stiffness. This progressive fibrogenic process leads to further fibrosis that gradually extends into the neighboring space [41].

NADPH plays a dual role in redox equilibrium. Glutathione, a cell’s low-molecular antioxidant, must be regenerated using NADPH; however, increasing NADPH levels activate NADPH oxidase (NOX), which generates oxidative stress, which causes cardiovascular diseases. NOX and other oxidative enzyme systems inactivate NO, which causes endothelial dysfunction by raising superoxide anion levels [35,42,43]. ROS and consequently oxidative stress may cause hypertension-related vascular alterations [41]. By monitoring NADP+/NADPH levels and dynamics, researchers can gain a deeper understanding of the complex connection between NADPH homeostasis, NOX enzyme activity, and the oxidative stress associated with cardiovascular illnesses. It is essential for the development of precise treatments that involve modulating NADP+/NADPH metabolism and redox balance [43]. Siwik et al. (2001) suggested that oxidative stress may cause myocardial remodeling [44]. In this study, the stressed group had higher total NADPH and NADPH/NADP+ ratio levels than the stress-naïve group. NADPH boosts cell antioxidant capability. It acts as a substrate for glutathione reductase, which results in GSSG reduction to GSH. Oxidative stress increases ROS generation and decreases antioxidant defense. Each organism adapts at the molecular and biochemical levels to oxidative stress, particularly chronic stress [45]; thus, we examined the gene expression of antioxidant proteins. Overexpression of *Sod2* and *Cat* may indicate a redox adaptive response. Free proline concentration increased considerably. Recent studies show that proline regulates biochemical processes [45,46,47]. The antioxidant capabilities of proline may explain the considerable rise in its concentration in response to oxidative stress in this study [47]. The findings suggest that proline synthesis regulates DNA proliferation and repair [46,47]. Summing up, we observed an induction of oxidative stress in the hearts of rats demonstrating depressive-like behavior. Oxidative stress may explain the link between cardiovascular problems and major depressive disorder [48]. These findings support Al et al. (2016), who reported oxidative stress and inflammation in CVS rats [49], and Greaney et al. (2019), who observed that oxidative stress causes microvascular dysfunction in patients with major depressive disorder [50]. We additionally identified antioxidant pathways activated in response to oxidative stress.

The heart muscle’s primary energy source is free FAs (FFAs), whose uptake into cells relies on their blood concentration and transporter density. FA transport protein (FATP), FA translocase (FAT/CD36), and FA-binding protein (FABP) are membrane transporters that deliver FA into cells. This study found upregulation of genes involved in FFA transport to cardiomyocytes and lipid synthesis. This is interesting as cardiomyocytes have a low FFA de novo biosynthesis capacity. However, this pathway may be activated under pathological conditions. Chronic FFAs uptake may cause cardiac insulin resistance [51]. Further, an increase in FFA oxidation is characteristic of the processes in the hearts of individuals with diabetes [52,53]. As previously mentioned, we demonstrated elevated NADPH concentration, which provides reducing equivalents for FA synthase to synthesize FA [54], and the NADPH/NADP^+^ ratio significantly contributes to the synthesis reactions. This may indicate another role for NADPH in the presented results. Increasing lipid oxidation was indicated by a significant drop in ethanolamine, which may be implicated in phospholipid synthesis. Activating FAs using acyl-CoA synthase creates long-chain FA and coenzyme A. *Acsl1*, the catalyst of FA metabolism, was upregulated in this study, suggesting enhanced FA metabolism.

There is evidence that an increase in NADPH concentration initiates the process of heart muscle remodeling. In turn, it may lead to impaired systolic and diastolic heart function and, in the final stage, to congestive heart failure [37]. We may observe altered gene expression, protein expression, and cardiomyocyte biology in the heart remodeling process. In this study, an altered proportion was found between the free isoform of the β (free, β-MHC) and α (fast, α-MHC) heavy chain, of which α-MHC is characteristic of the fetal heart. The molecular analysis showed *Myh6* upregulation and *Myh7* downregulation, which is typical of the embryonic period and reduces heart muscle contractility [38,40,55,56]. Some studies link the increase in *Myh6* expression with cardiomyopathy [57], and gene mutations may cause coronary heart disease [58]. We also found elevated NT-proBNP levels, which increase systolic and diastolic dysfunction and left ventricular hypertrophy. Its determination is beneficial in the initial stage of the diagnosis of heart failure in determining the prognosis in patients with heart failure and in assessing the impact and effectiveness of heart failure treatment [59,60,61]. In this study, NT-proBNP levels were significantly elevated, possibly as a response to overload, suggesting the possibility of an onset of the heart remodeling process. This thesis is consistent with the results obtained for the ratio of the heart mass to the body mass of rats in the study group, which was significantly higher than in the control group and may indicate remodeling-induced hypertrophy [55,56]. Finally, significantly lower median body weight indicates an adverse systemic effect of the CVS procedure accompanying depressive-like behavior [62]. A high heart-to-total body weight ratio may characterize the hypertrophy accompanying myocardial remodeling.

This study has some limitations. We acknowledge that using a single animal model is not useful for mimicking mental diseases. The CVS model, which replicates the core depressive symptom of anhedonia, is essential in pre-clinical research on depression in this particular context. Despite its limitations, the CVS paradigm has been effective in drug development and interdisciplinary research for gaining insight into the neuroscience of depression.

## 4. Materials and Methods

### 4.1. Animals

All procedures were carried out by mandatory European standards for conducting experimental studies on animal models (European Communities Council Directive of 22 September 2010 [2010/63/EU]) and Polish legislation acts concerning animal experimentation. The consent to conduct animal experiments was given by the Local Ethical Committee for Animal Experiments (approval number 12/2015).

The experimental procedures were carried out using twenty male Wistar rats (Han IGS Rat [Crl:WI(Han)]) aged ten weeks with an average body weight of 250 ± 15 g in the Centre of Experimental Medicine, Medical University of Lublin, Poland. The animals were housed in colony cages at room temperature (22 ± 3 °C) with standard feed during the experiment and had constant access to water unless the procedure schedule required otherwise (CVS procedure). The normal day-light cycle (6 a.m. to 6 p.m., day; 6 p.m. to 6:00 a.m., night) was maintained in the brewery room.

### 4.2. CVS Procedure

After seven days of acclimatization, the animals were randomly divided into the control group (CTL) and rats subjected to CVS procedures (CMS, to distinguish the group name from the procedure name). Each group consisted of 10 individuals. CTL rats were kept free without being subjected to any procedures, while rats in the CMS group were exposed to various stress conditions for 40 days [1]. The following stressors were used in the experiment: 24 h of food or water restriction, 1 hour of immobilization in a sealed plastic tube (26 × 6 cm), 2 h restraint at 4 °C, forced swimming for 10 min, blinker for 3 h (rats were placed in open-field container made of brown plywood with a glass front wall; a 40 W flash with 60 flashes per minute was used) and 24-hour-isolation (Figure 9).

To avoid predictability, rats were exposed to stressors at different times each day. After 40-day stress procedures, the rats were sacrificed by decapitation, which is one of the acceptable methods of rodent euthanasia [63,64]. The weight of the rats was measured over 40 days. The measurement time was limited by the planned sacrifice of the animals after the experiment. Rats’ hearts were collected for examination, rinsed with saline, dried, weighed, and frozen at −80 °C. The blood was also collected for biochemical tests. The serum was separated and frozen at −80 °C.

### 4.3. Behavior

To confirm the depressive-like behavior induced by the CVS protocol, the forced swim test was performed. The procedure was conducted according to Porsolt et al. as the last stressor of the CVS protocol [65]. Briefly, the rats were placed twice in a glass cylinder (40 cm height, 25 cm diameter) containing 30 cm of water (22.5 ± 0.5 °C) for a 15 min pretest. The rats were then dried and removed from their cages. Twenty-four hours later, the rats were exposed to the same experimental conditions outlined above for 5 min when the immobility time was recorded.

### 4.4. Biochemical Assays

Concentrations of nicotinamide adenine dinucleotide (NAD^+^) and its reduced form (NADH) were measured in collected tissue using an NAD/NADH Assay Kit (BioChain, San Francisco Bay Area, CA, USA) according to the manufacturer’s protocol. Briefly, 20 mg of tissue was washed with cold saline (PBS, Phosphate Buffered Saline; 1X, pH 7.4) and then homogenized with an extraction buffer for NAD^+^ or NADH. The extracts were incubated at 60 °C for 5 min with the appropriate buffer. A working reagent of test buffer, enzymes, lactate, and tetrazole was added. The absorbance was measured at λ = 565 nm at T_0_ and T_15_. NADP+ and NADPH concentrations were measured in 20 mg of collected tissue using an NADP/NADPH Assay Kit (BioChain, San Francisco Bay Area, CA, USA) according to the manufacturer’s protocol. An absorbance was measured at λ = 565 nm at T_0_ and T_15_. The analysis was performed in three technical repetitions.

The NT-proBNP, Ldha, and Ldhb protein content was performed using the ELISA method according to the manufacturer’s instructions (Clontech Laboratories, Mountain View, CA, USA). For this purpose, total protein was isolated from heart homogenates using a lysis buffer. Color changes were measured spectrophotometrically at a wavelength λ = 450 nm, and NT-proBNP concentrations were determined in relation to standard curves. The data are presented as the content per 1 mg of total protein (*n* = 10 trials from 10 animals per group).

The serum fasting glucose and lactate concentrations were measured using disposable test strips for measuring glucose (GlucCell Glucose Monitoring System, CESCO Bioengineering, Taichung, Taiwan) and lactate (Lactate Plus Meter, Nova Biomedical, Waltham, MA, USA) (*n* = 10 trials from 10 animals per group). The result for fasting glucose is given in mg/dL; for lactate in mmol/L.

### 4.5. Free Amino Acid Analysis

Free amino acid analysis was performed using the AAA 500 amino acid analyzer (Ingos, Prague, Czech Republic). A total of 50 mg of tissue was weighed and then sonicated (Ultrasonic homogenizer JY 92-IIN1, NingBo Scientz Biotechnology, Ningbo, China) (total sonication time: 2 min; sonication time: 5 s, interval time: 5 s; power: 40%) and deproteinized with 6% sulfosalicylic acid in lithium citrate buffer (pH 2.8 in a 1:10 ratio), followed by centrifugation using a centrifuge (Eppendorf, Hamburg, Germany). The supernatant was used to assess the level of free amino acids by ion exchange chromatography using five lithium citrate buffers (pH 2.60; 3.10; 3.35; 4.05; and 4.65). Amino acids were separated on an Ostion LG FA analytical column and identified using INGOS standards using Clarity v.6.1.00.130 software (DataApex, Czech Republic). Data were divided into metabolic pathways and described as the mean and its standard deviation, as well as statistical significance and fold change for the median in relation to the control group (*n* = 10 trials from 10 animals per group).

### 4.6. Oxidative DNA Damage

DNA was isolated with the Syngen DNA Mini Kit (Syngen, Wrocław, Poland) according to the manufacturer’s instructions. The concentration of the genomic DNA was measured using the MaestroNano Micro-Volume Spectrophotometer (Maestrogen Inc., Hsinchu, Taiwan) and adjusted to 100 ng/µL in the TE buffer. The number of AP sites was evaluated using the DNA Damage Quantification Kit (Dojindo, Kumamoto, Japan) according to the manufacturer’s protocol. The method is based on a specific reaction of an aldehyde-reactive probe (ARP; N′-aminooxymethylcarbonylhydrazin-D-biotin) with an aldehyde group present on the open-ring form of AP sites. AP sites were tagged with biotin residues and were quantified using avidin-biotin assay followed by colorimetric detection of peroxidize conjugated to the avidin at 650 nm using a PowerWave microplate spectrophotometer (BioTek Instruments, Winooski, VT, USA). The detection limit of the number of abasic sites is 0–40 per 1 × 10^5^ bp.

### 4.7. Molecular Analysis

RNA was isolated using the TRIzol reagent (Invitrogen, Waltham, MA, USA) and the Chomczynski and Sacchi method [66]. Briefly, 50 mg tissue sections from the left ventricle were homogenized in the presence of TRIzol reagent, after which chloroform was added (POCH, Gliwice, Poland). The tube was centrifuged to separate the phases of the aqueous (containing RNA), interphase (containing DNA), and organic phase (containing proteins). Then, RNA was precipitated from the aqueous phase with isopropanol (POCH, Gliwice, Poland). The precipitated RNA was rinsed with 75% ethanol (POCH, Gliwice, Poland), then centrifuged and dissolved in RNase-free water (EURx, Gdańsk, Poland). Then, the RNA concentration and purity were measured using a MaestroNano micro-volume spectrophotometer (Maestrogen, Hsinchu, Taiwan). High-purity RNA (RNA: A260/280 > 1.9) was used for the cDNA synthesis. RNA concentrations were normalized, and the reverse transcription was performed using the NG dART RT-PCR kit (EURx, Gdańsk, Poland) and random hexamers according to the manufacturer’s instructions to obtain cDNA. The reaction was carried out using a thermocycler (Mastercycler gradient, Eppendorf, Hamburg, Germany) and the following thermal profile: 10 min at 25 °C, 50 min at 50 °C, 5 min at 80 °C. Each reaction was performed in triplicates.

To analyze changes occurring under the influence of CMS, the primers were designed using the nucleotide sequence database (NCBI) and the Primer3 application (v.0.4.0). Nucleotide synthesis was performed by Genomed (Warsaw, Poland). The sequences of used primers are presented in Table 4. 

The relative difference in expression levels between the control and the test sample is a sensitive mRNA quantification method that requires the constant expression of selected reference genes. The following reference genes were chosen based on Brattelid et al.’s [67] analysis, ribosomal protein 32 (*Rpl32*) and RNA polymerase II subunit A (*Polr2a*), using the requirements of MIQE Guidelines [68].

The analysis of the genes’ expression levels was performed by real-time PCR using the high-throughput SmartChip MyDesign Chip system (WaferGen Bio-Systems, Fremont, CA, USA). The procedures were carried out in accordance with the manufacturer’s protocols with minor modifications. The sample reaction mix consisted of PowerUp SYBR Green Master Mix (Applied Biosystems, Foster, CA, USA), RNase-free water, and cDNA. The samples were transferred to SmartChip by Multisample Nanodispenser. Subsequently, the assay reagent mix consisting of PowerUp SYBR Green Master Mix (2×), RNase-free water, and oligonucleotides was transferred by the automatic dispenser to the SmartChip. When the transfer was completed, the chip was sealed, centrifuged, and placed in the SmartChip Cycler. Each reaction was carried out in four technical replications (*n* = 40 trials from 10 animals per group). The reaction profile was as follows: 95 °C for 2 min; 45 cycles, 95 °C for 15 s; 57 °C for 15 s; and 72 °C for 1 min—melt curve 0.4 °C/s up to 97 °C. The quality of the samples was analyzed based on the amplification values, Tm and Ct, to remove any deviations before calculating ΔΔCt and to determine the fold change in mRNA levels. Data on changes in gene expression at the mRNA level were presented as mean RQ ± SD.

### 4.8. Histological Staining

The material was collected from the left ventricle of each individual in buffered 10% formalin and processed to paraffin blocks. Four-micrometer slides were cut on the microtome and stained with hematoxylin and eosin, picric acid, and acid fuchsin (van Gieson’s staining), as well as periodic acid and fuchsine (PAS, Periodic acid–Schiff), and PAS with diastase. An experienced blind pathologist examined the slides with a light microscope. Two distinct sections of each staining were assessed in each animal from a single heart. Histological changes were classified as follows: “−” for no changes, “+” for minor changes, “++” for moderate alterations, and “+++” for significant changes.

### 4.9. Statistical Analysis

Statistical analysis was performed using the STATISTICA v.10 application (StatSoft, Cracow, Poland) and GraphPad Prism v.5.0, www.graphpad.com (accessed on 29 December 2023 and 5 January 2024). The results are presented as mean ± standard deviation (SEM). The normality of the distribution was analyzed using the Shapiro–Wilk test. The comparison between study groups was analyzed with a Student’s *t*-test or Mann–Whitney U test. The results were presented as statistically significant when *p* ≤ 0.05.

## 5. Conclusions

This study provides direct evidence that prolonged stress induces depressive-like behavior in rats and reprogrammes heart muscle at the molecular, biochemical, and tissue levels. The alterations include myocardial metabolism—such as the redirection of glycolysis to the synthesis of glycogen and the activation of the PP pathway—disturbances in myocardial amino acid metabolism, oxidative DNA damage, and significant changes in gene expression associated with glucose metabolism, oxidative stress, and cardiac remodeling pathways. Transcriptional signals for anabolism, catabolism, lipid transport, and the gluconeogenesis pathway with glycogen storage indicate an increase in ATP uptake from lipid metabolism, not glucose. Moreover, histological staining revealed minor morphological changes, including increased acidophilicity of cells, collagen deposition surrounding blood vessels, and glycogen accumulation. Finally, stress-induced gene expression alterations and NT-ProBNP increases may signal myocardial remodeling in stressed rats. Our findings may be used to explain the common mechanisms of depression and cardiovascular diseases in people exposed to chronic stress, and may also contribute to the development of innovative approaches to the prevention and treatment of these diseases. In the event of cardiac abnormalities, the promising target of heart disease treatment is inhibiting FA oxidation or increasing glucose oxidation, which will support normal myocardial function.

Future research should further elucidate the molecular pathways linking chronic stress to myocardial reprogramming. Key areas include detailed studies on the redirection of glycolysis, the activation of the pentose phosphate pathway, and the disturbances in myocardial amino acid metabolism. Exploring the potential of targeting FA oxidation and promoting glucose oxidation as therapeutic strategies for heart disease in the context of chronic stress may be viable. Developing and testing specific inhibitors of FA oxidation or agents that enhance glucose oxidation could lead to new treatments. Open perspectives involve understanding the broader implications of these findings for human health, particularly the common mechanisms underlying depression and cardiovascular diseases. This knowledge could contribute to innovative preventive and therapeutic approaches for individuals exposed to chronic stress, ultimately improving mental and cardiovascular health outcomes. Collaborative studies integrating clinical and preclinical research will be essential to translate these findings into effective interventions.

## Figures and Tables

**Figure 1 ijms-25-05899-f001:**
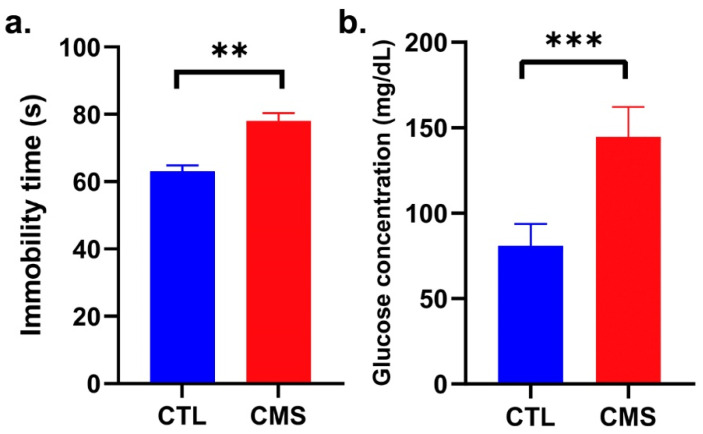
Effect of chronic variable mild stress on (**a**) immobility time in a forced swimming test; (**b**) fasting glucose levels. Data are expressed as means ± standard deviation (*n* = 10). A Student’s *t*-test was used. *** *p* ≤ 0.001, ** *p* ≤ 0.01 compared to the control group. CTL, control group; CMS, chronic variable mild stress group.

**Figure 2 ijms-25-05899-f002:**
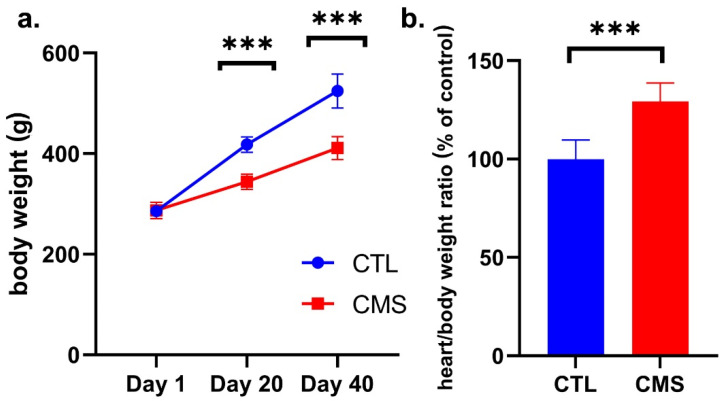
(**a**) Effect of chronic variable mild stress procedure on rats’ body weight; (**b**) the heart/body weight ratio of rats subjected to chronic variable mild stress procedure. Data are expressed as means ± standard deviation (*n* = 10). A Student’s *t*-test was used. *** *p* ≤ 0.001 compared to the control group. CTL, control group; CMS, chronic variable mild stress group.

**Figure 3 ijms-25-05899-f003:**
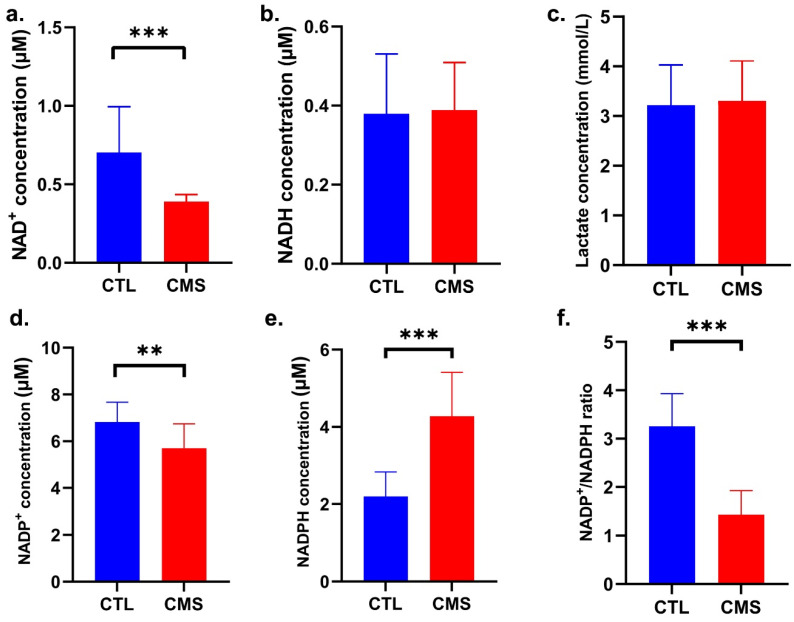
Effect of CVS procedure on biochemical parameters: (**a**) NAD+; (**b**) NADH; (**c**) lactate; (**d**) NADP+; (**e**) NADPH concentrations; (**f**) NADP+-to-NADPH ratio. Data are expressed as means ± standard deviation (*n* = 10). ** *p* ≤ 0.01; *** *p* ≤ 0.001 compared to the control group. CTL, control group; CMS, chronic variable mild stress group.

**Figure 4 ijms-25-05899-f004:**
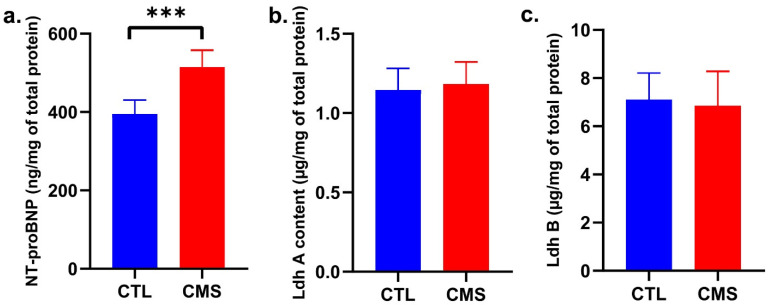
Effect of procedure on biochemical parameters: (**a**) NT-proBNP; (**b**) Ldh A; (**c**) Ldh B content. Data are expressed as means ± standard deviation (*n* = 10). *** *p* ≤ 0.001 compared to the control group. CTL, control group; CMS, chronic variable mild stress group.

**Figure 5 ijms-25-05899-f005:**
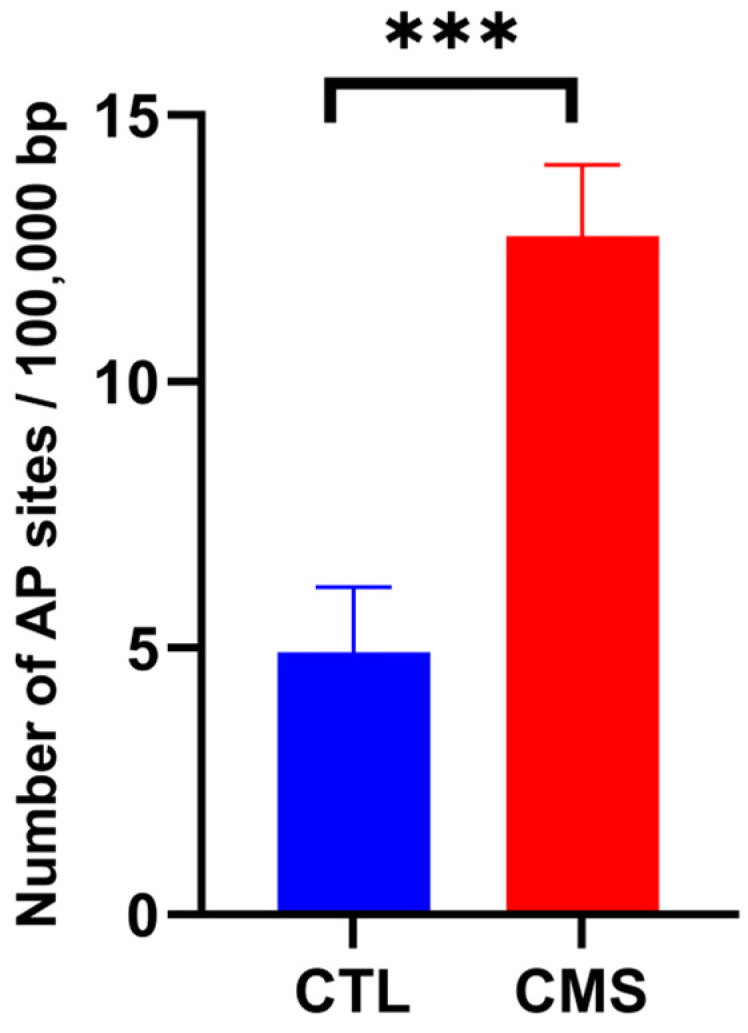
Effect of chronic variable mild stress on a number of abasic sites per 100,000 bp in DNA of rats subjected to the procedure. Data are expressed as means ± standard deviation (*n* = 10). A Student’s *t*-test was used. *** *p* ≤ 0.001 compared to the control group. CTL, control group; CMS, chronic variable mild stress group.

**Figure 6 ijms-25-05899-f006:**
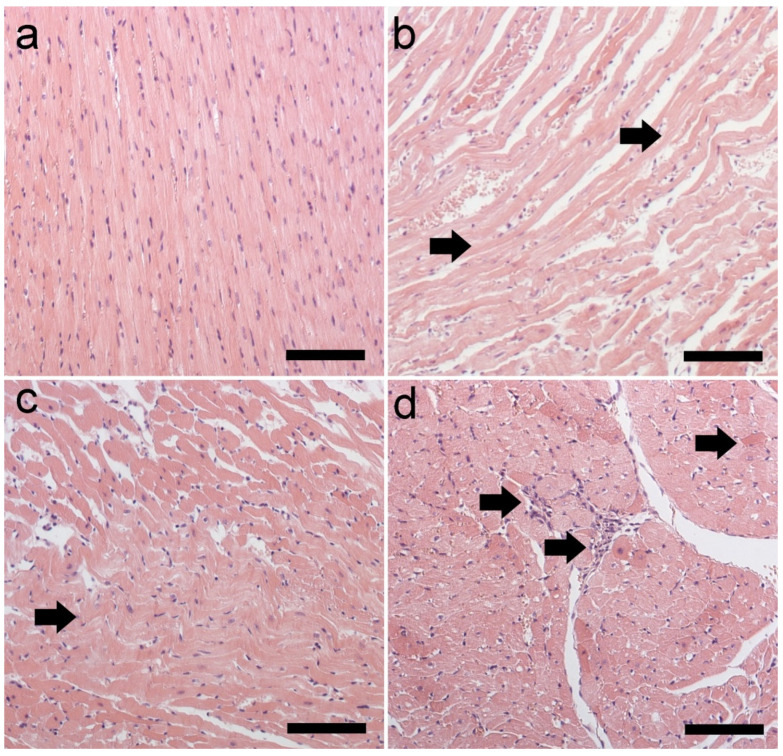
Section of rat heart muscle. H + E staining. Magnification ×150 (scale bar: 150 μm). (**a**) Normal histopathology picture of the rat heart in the CTL group; (**b**,**c**) wavy course of fibers (→) indicating necrosis in the CMS group; (**d**) mononuclear cell infiltration (→) and increased acidophilus of individual cardiomyocytes (→) in the CMS group. CTL, control group; CMS, study group.

**Figure 7 ijms-25-05899-f007:**
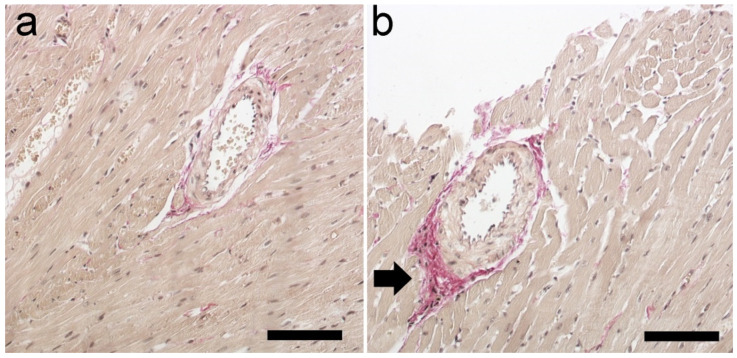
Section of rat heart muscle. Van Gieson’s staining. Magnification ×150 (scale bar: 150 μm). (**a**) Normal degree of collagen surrounding an arteriole in the CTL group; (**b**) collagen more intensely stained around the arteriole extending into the surrounding myocardium (→) in the CMS group. CTL, control group; CMS, study group.

**Figure 8 ijms-25-05899-f008:**
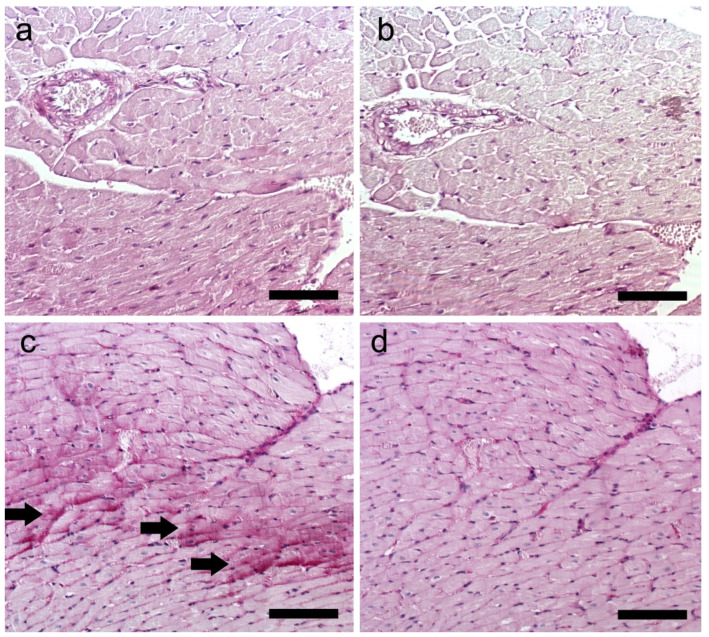
Section of rat heart muscle. Magnification ×150 (scale bar: 150 μm). (**a**) PAS staining in the CTL group; (**b**) PAS with diastase in the CTL group as a control staining; (**c**) PAS staining in the CMS group, (→) a visible accumulation of glycogen; (**d**) PAS with diastase in CMS group. CTL, control group; CMS, study group.

**Figure 9 ijms-25-05899-f009:**
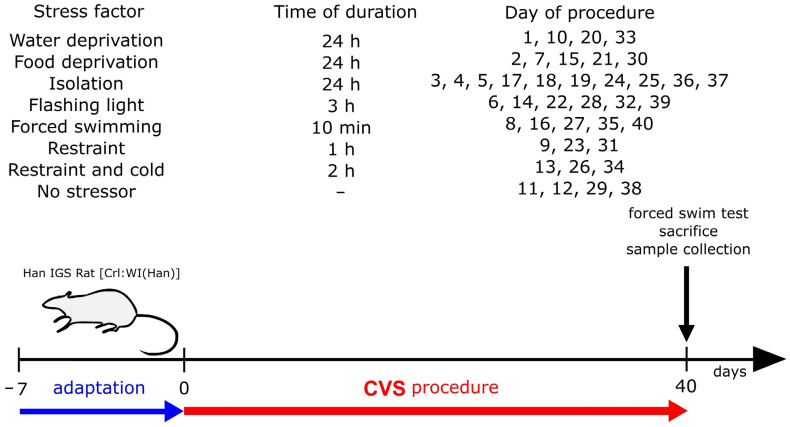
Schematic representation of chronic variable mild stress experimental protocol. CVS, chronic variable mild stress.

**Table 1 ijms-25-05899-t001:** The results of free amino acid concentration analysis in heart homogenates of rats subjected to chronic variable mild stress procedure. Data are presented as median concentration (µmol/mL) and its standard deviation. Biochemical names of amino acids are assigned to pathways. The *p*-values were calculated with the Mann–Whitney U test.

Pathway	Biochemical Name	CTL	CMS	Fold Change	*p*-Value
		Median Concentration ± SD [µmol/mL]			
Glycine, serine, and threonine metabolism	glycine	2.852 ± 0.863	3.527 ± 0.582	1.237	0.135
	serine	3.265 ± 1.147	2.644 ± 0.652	0.810	0.285
	phosphoserine	0.734 ± 0.043	0.663 ± 0.059	0.903	0.031
	threonine	18.774 ± 6.903	22.289 ± 5.210	1.187	0.362
Alanine and aspartate metabolism	alanine	16.282 ± 2.097	17.534 ± 3.611	1.077	0.388
	asparagine	4.360 ± 1.851	7.842 ± 2.160	1.799	0.011
Glutamate metabolism	glutamine	42.344 ± 10.766	38.485 ± 8.400	0.909	0.506
	glutamic acid	68.252 ± 13.054	80.088 ± 14.123	1.173	0.095
	gamma-aminobutyrate (GABA)	0.074 ± 0.021	0.060 ± 0.022	0.811	0.302
Histidine metabolism	histidine	0.832 ± 0.084	0.901 ± 0.215	1.083	0.615
	1-methylhistidine	0.194 ± 0.068	0.202 ± 0.032	1.041	0.615
Lysine metabolism	lysine	1.728 ± 0.141	1.641 ± 0.324	0.950	0.468
Phenylalanine and tyrosine metabolism	phenylalanine	0.836 ± 0.137	0.882 ± 0.129	1.055	0.621
	tyrosine	0.660 ± 0.061	0.593 ± 0.064	0.898	0.082
Valine, leucine, and isoleucine metabolism	isoleucine	0.632 ± 0.136	0.738 ± 0.089	1.168	0.074
	leucine	1.369 ± 0.197	1.406 ± 0.150	1.027	0.799
	valine	1.081 ± 0.264	1.559 ± 0.191	1.442	0.003
Cysteine, methionine, SAM, and taurine metabolism	cysteine	0.069 ± 0.013	0.079 ± 0.010	1.145	0.151
	taurine	111.129 ± 5.822	106.945 ± 7.418	0.962	0.271
	methionine	0.520 ± 0.053	0.663 ± 0.078	1.275	0.003
Urea cycle; arginine and proline metabolism	arginine	1.780 ± 0.243	1.380 ± 0.274	0.775	0.013
	ornithine	0.036 ± 0.016	0.038 ± 0.008	1.056	0.799
	urea	28.327 ± 3.093	28.470 ± 3.375	1.005	0.815
	proline	0.161 ± 0.025	0.248 ± 0.102	1.540	0.008
	citrulline	0.361 ± 0.089	0.302 ± 0.056	0.837	0.146
Glycerolipid metabolism	ethanolamine	0.249 ± 0.028	0.191 ± 0.035	0.767	0.006

SD, standard deviation; CTL, control group; CMS, chronic variable mild stress group; *p*-value, statistical significance.

**Table 2 ijms-25-05899-t002:** The results of mRNA gene expression. Significantly changed median RQ levels are marked with a color scale. *p*-value was calculated using the Mann–Whitney U test.

Pathway	Gene Symbol	CTL	CMS	*p*-Value		
		Me RQ ± SD	Me RQ ± SD			
Glucose transport	*Slc2a1*	1.010 ± 0.145	0.993 ± 0.160	0.776	Scale (RQ)	
	*Slc2a4*	1.008 ± 0.131	1.193 ± 0.146	<0.001		>1.6
Glucose metabolism	*Hk2*	1.017 ± 0.182	1.417 ± 0.220	<0.001		1.21–1.59
	*Pfkl*	1.005 ± 0.097	1.036 ± 0.176	0.538		0.8–1.20
	*Tpi1*	1.008 ± 0.130	1.182 ± 0.148	<0.001		0.5–0.79
	*Got2*	1.011 ± 0.155	1.117 ± 0.203	0.006		0–0.49
Gluconeogenesis	*Fbp1*	1.015 ± 0.174	0.441 ± 0.151	<0.001		
	*Fbp2*	1.009 ± 0.136	2.116 ± 1.521	<0.001		
	*G6pc*	1.008 ± 0.131	1.642 ± 0.558	<0.001		
	*Pc*	1.012 ± 0.156	1.341 ± 0.730	0.714		
Glutaminolysis	*Slc1a5*	1.011 ± 0.149	1.441 ± 0.245	<0.001		
	*Gls*	1.019 ± 0.200	0.954 ± 0.138	0.208		
Pyruvate metabolism	*Me1*	1.012 ± 0.158	1.161 ± 0.279	0.021		
	*Ldha*	1.005 ± 0.104	1.273 ± 0.120	<0.001		
	*Pdk2*	1.006 ± 0.117	1.010 ± 0.146	0.823		
	*Pdk4*	1.014 ± 0.170	4.523 ± 1.753	<0.001		
	*Mpc1*	1.005 ± 0.107	1.038 ± 0.160	0.396		
	*Mpc2*	1.007 ± 0.118	1.085 ± 0.202	0.155		
Monocarboxylate transport	*Slc16a1*	1.006 ± 0.113	1.471 ± 0.231	<0.001		
Urea cycle	*Cs*	1.007 ± 0.122	1.237 ± 0.186	<0.001		
	*Sdhc*	1.005 ± 0.106	1.045 ± 0.094	0.118		
	*Mdh1*	1.006 ± 0.114	1.178 ± 0.208	0.001		
	*Idh2*	1.010 ± 0.150	1.152 ± 0.144	0.001		
Lipid transport	*Cd36*	1.004 ± 0.093	1.319 ± 0.108	<0.001		
	*Slc27a1*	1.011 ± 0.145	1.231 ± 0.205	<0.001		
	*Cpt2*	1.011 ± 0.154	1.005 ± 0.184	0.762		
Lipid synthesis	*Acaca*	1.014 ± 0.172	1.058 ± 0.243	0.653		
	*Dgat1*	1.005 ± 0.100	1.255 ± 0.323	0.004		
	*Acly*	1.016 ± 0.185	1.286 ± 0.180	<0.001		
	*Prkaa2*	1.012 ± 0.162	1.555 ± 0.249	<0.001		
	*Mlycd*	1.005 ± 0.107	1.101 ± 0.236	0.304		
β-oxidation	*Cpt1a*	1.007 ± 0.123	1.581 ± 0.0256	<0.001		
	*Cpt1b*	1.014 ± 0.171	1.030 ± 0.170	0.890		
	*Ppargc1a*	1.012 ± 0.161	1.942 ± 0.324	<0.001		
	*Ppara*	1.008 ± 0.130	1.002 ± 0.256	0.728		
	*Acsl1*	1.011 ± 0.155	1.218 ± 0.230	0.001		
	*Acadm*	1.014 ± 0.174	1.202 ± 0.228	0.002		
Oxidative stress	*Cat*	1.013 ± 0.164	1.216 ± 0.241	0.002		
	*Sod2*	1.009 ± 0.135	1.200 ± 0.181	<0.001		
Heart remodeling	*Myh6*	1.012 ± 0.160	1.663 ± 0.316	<0.001		
	*Myh7*	1.017 ± 0.192	0.628 ± 0.203	<0.001		
Transcription factors	*Myc*	1.016 ± 0.186	1.904 ± 0.244	<0.001		
	*Hif1a*	1.008 ± 0.128	1.150 ± 0.200	0.005		
	*Srebp1c*	1.008 ± 0.129	0.975 ± 0.164	0.313		

CTL, control group; CMS, chronic variable mild stress group; Me, mean; *p*-value, statistical significance; SD, standard deviation.

**Table 3 ijms-25-05899-t003:** The presence and intensity of morphological changes in rat hearts after chronic variable mild stress procedure.

Morphological Feature	CTL	CMS
Mononuclear cell infiltration	− (8)	+ (8)
Wavy course of fibers (necrosis)	− (9)	+ (9)
Increased acidophilus (necrosis)	− (10)	+ (9)
Collagen deposition	− (9)	+ (10)
Glycogen deposition	− (10)	++ (10)

−, no changes; +, changes of minor intensity; ++, moderate alterations; incidence in the group was given in brackets (*n* = 10); CTL, control group; CMS, study group.

**Table 4 ijms-25-05899-t004:** The oligonucleotide sequences of used primers divided into pathways/functions. Gene symbols are also given.

Pathway/Function	Gene Symbol	RefSeq	Sequence (5′ → 3′)	
Glucose transport	*Slc2a1*	NM_138827.1	F: gcc tga gac cag ttg aaa gc	R: gag tgt ccg tgt ctt cag ca
	*Slc2a4*	NM_012751.1	F: gct tct gtt gcc ctt ctg tc	R: tgg acg ctc tct ttc caa ct
Glucose metabolism	*Hk2*	NM_012735.2	F: ctc cat ccc aca gga ggt ta	R: tga gga gga tgc tct ggt ct
	*Pfkl*	NM_031715.1	F: ggc ttt gag gct tac aca gg	R: cag act gct tga ttc ggt ca
	*Tpi1*	NM_022922.2	F: atg tca cca aga tgg ctt cc	R: aca cat cca gta ggg ctt gg
	*Got2*	NM_013177.2	F: acc atc cac tgc cgt ctt ac	R: tct tga agg ctt cgg tca ct
Gluconeogenesis	*Fbp1*	NM_012558.3	F: tta tag gct ccc cca gga ct	R: cag ggt gct gat atc cgt tt
	*Fbp2*	NM_053716.1	F: cag ggg atg agg tga aga aa	R: agt ggg tca aag caa acc ac
	*G6pc*	NM_013098.2	F: acc ctg gta gcc ctg tct tt	R: ggg ctt tct ctt ctg tgt cg
	*Pc*	NM_012744.2	F: gag att gcc atc cga gtg tt	R: ctc ctt ggc cac ctt aat ga
Glutaminolysis	*Slc1a5*	NM_175758.3	F: ttc tat gcc ctg gtg acc tc	R: tgc cat ctc ctt gaa tag gg
	*Gls*	NM_012569.2	F: cac aca cac gga ttt ctt gg	R: gcc gaa gct gac ttt gaa ac
Pyruvate transport	*Mpc1*	NM_133561.1	F: act ttc gcc ctc tgt tgc ta	R: gca ctg tcc ctt tca aga gc
	*Mpc2*	NM_001077643.2	F: ttt tct ggg ctc cga taa tg	R: gat ccg aaa cag ctg aga gg
Pyruvate metabolism	*Me1*	NM_012600.2	F: gcc ctg aat atg atg cgt tt	R: cct gga aca gca ctg tct ga
	*Ldha*	NM_017025.1	F: ggt ggt tga cag tgc ata cg	R: agg ata cat ggg acg ctg ag
	*Pdk2*	NM_030872.1	F: agg aag tca atg cca cca ac	R: ttt tga tgg gag gga gag tg
	*Pdk4*	NM_053551.1	F: cct ttg gct ggt ttt ggt ta	R: cac cag tca tca gcc tca ga
Monocarboxylate transport	*Slc16a1*	NM_012716.2	F: tat gcc gga ggt cct atc ag	R: agt tga aag caa gcc caa ga
Urea cycle	*Cs*	NM_130755.1	F: aag gct aaa ggt ggg gaa ga	R: gtg cag att ggt cgg aaa at
	*Sdhc*	NM_001005534.1	F: ctt tgt gct gct tct tgc tg	R: cca cgc tgc tct tta tct cc
	*Mdh1*	NM_033235.2	F: gaa gcc ctc aaa gac gac ag	R: cga cag gga acg agt aga gc
	*Idh2*	NM_001014161.1	F: cag tct gac atc ctg gct ca	R: aga tgc tgg caa tag ggt tg
Lipid transport	*Cd36*	NM_031561.2	F: gca aca aca agg cca ggt at	R: aag agc tag gca gca tgg aa
	*Slc27a1*	NM_053580.2	F: cct cac atc aca gca gga ga	R: gct ctg tcc aca ccc ttc at
	*Cpt2*	NM_012930.1	F: tcc tcg atc aag atg gga ac	R: gat cct tca tcg gga agt ca
Lipid synthesis	*Acaca*	NM_022193.1	F: tac aac gca ggc atc aga ag	R: tgt gct gca gga aga ttg ac
	*Dgat1*	NM_053437.1	F: ctg ctg ccc aga aca ctg ta	R: aag ctg ggt gaa aaa gag ca
	*Acly*	NM_016987.2	F: ctc aca cgg aag ctc atc aa	R: atg gca aca ccc tcg tag ac
	*Prkaa2*	NM_023991.1	F: agc tcg cag tgg ctt atc at	R: ggg gct gtc tgc tat gag ag
	*Mlycd*	NM_053477.1	F: gcc tgg tac ctt tac ggt ga	R: gct acc agg ctg agg atc tg
β-oxidation	*Cpt1a*	NM_031559.2	F: atg acg gct atg gtg tct cc	R: gtg agg cca aac aag gtg at
	*Cpt1b*	NM_013200.1	F: gca aac tgg acc gag aag ag	R: cct tga aga agc gac ctt tg
	*Ppargc1a*	NM_031347.1	F: atg tgt cgc ctt ctt gct ct	R: atc tac tgc ctg ggg acc tt
	*Ppara*	NM_013196.1	F: tca cac aat gca atc cgt tt	R: ggc ctt gac ctt gtt cat gt
	*Acsl1*	NM_012820.1	F: aac gat gta cga tgg ctt cc	R: ggt cac cca ctc agg tct gt
	*Acadm*	NM_016986.2	F: caa gag agc ctg gga act tg	R: ccc caa aga att tgc ttc aa
Oxidative stress	*Cat*	NM_012520.2	F: aca tgg tct ggg act tct gg	R: caa gtt ttt gat gcc ctg gt
	*Sod2*	NM_017051.2	F: cac tgt ggc tga gct gtt gt	R: tcc aag caa ttc aag cct ct
Heart remodeling	*Myh6*	NM_017239.2	F: tga tga ctc cga gga gct tt	R: tga cac aga ccc ttg agc ag
	*Myh7*	NM_017240.2	F: cct cgc aat atc aag gga aa	R: tac agg tgc atc agc tcc ag
Transcription factors	*Myc*	NM_012603.2	F: cga gct gaa gcg tag ctt tt	R: ctc gcc gtt tcc tca gta ag
	*Hif1a*	NM_024359.1	F: tca agt cag caa cgt gga ag	R: tat cga ggc tgt gtc gac tg
	*Srebp1c*	NM_001276707.1	F: gtg gtc ttc cag agg ctg ag	R: ggg tga gag cct tga gac ag
Reference genes	*Rpl32*	NM_013226	F: aga ttc aag ggc cag atc ct	R: cga tgg ctt ttc ggt tct ta
	*Polr2a*	XM_343922	F: cgt atc cgc atc atg aac agt ga	R: tca tcc atc tta tcc acc acc tct t

## Data Availability

The data presented in this study are available on request from the corresponding author.

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
