# Peer review of "Pathological Changes and Metabolic Adaptation in the Myocardium of Rats in Response to Chronic Variable Mild Stress"

_ijms, 2024, doi:10.3390/ijms25115899_

Round 1

Reviewer 1 Report

Comments and Suggestions for Authors

The manuscript titled “Pathological changes and metabolic adaptation in the myocardium of rats in response to chronic variable mild stress” by Ostrowska-Lésko, M.; et al. is a scientific work where the authors monitored the impact of mild stress factors on the myocardial metabolism and how these changes can eventually lead to heart diseases. The most relevant outcomes found by the authors could open new gates in the design of the next-generation of smart therapies to fight against heart malignancies. The manuscript is generally well-written and this is a topic of growing interest.

However, it exists some points that need to be addressed (please, see them below detailed point-by-point) to improve the scientifc quality of the submitted manuscript paper before this article will be consider for its publication in the International Journal of Molecular Sciences.

I) KEYWORDS. The authors should consider to add the term “oxidative stress” in the keyword list.

2) INTRODUCTION. “Chronic variable mild stress (CVS) (…) depressive-like behavior (…) progression of cardiovascular diseases” (lines 30-40). Could the authors provide some quantitative insights according to the worldwide burdens of cardiovascular diseases? This will greatly aid the potential reader to better understand the significance to conduct this research.

3) RESULTS. “2.2. Body weight assessment” (lines 86-95). Why did the authors only monitor the first 40 days after chronic variable mild stress exposure and not longer time scales? (Figure 9 in line 349 indicates that the rats are forced to die after these 40 days). A brief statement should be furnished in this regard.

4) “2.5. Oxidative DNA damage” (lines 133-143). What is the detection limit of the methodology followed by the authors to detect the number of DNA abasic sites? Some information should be provided in this regard.

5) Figure 6 (line 166). The lateral scale bar should be added to the histological section images. Same comment for the Fig. 7 (line 174), and Fig. 8 (line 181).

6) DISCUSSION. This section is well-structured. “NADPH plays a dual role in redox equilibrium (…) increase NADPH levels activate NADPH oxidase  (…) which causes endothelial dysfunction by raising superoxide anion levels (…) Each organism adapts at the molecular and biochemical levels to oxidative stress” (lines 257-268). I agree with the authors according the information stated in these statements albeit it may be also desirable to indicate how the detection of molecular levels of NADP+/NADPH [1] is crucial to better understand the inner cellular homeostasis with NADPH Oxidase enzyme which can lead to cardiovascular malignancies [2].

[1] Pérez-Domínguez, S.; et al. Nanomechanical Study of Enzyme: Coenzyme Complexes: Bipartite Sites in Plastidic Ferredoxin-NADP+ Reductase for the Interaction with NADP. Antioxidants 2022, 11, 537. https://doi.org/10.3390/antiox11030537.

[2] Poznyak, A.V.; et al. NADPH Oxidases and Their Role in Atherosclerosis. Biomedicines 2020, 8, 206. https://doi.org/10.3390/biomedicines8070206.

7) MATERIALS & METHODS. The authors perfectly explained all the steps “The supernatant (…) (pH 2.6; 3.1; 3.35; 4.05 and 4.65)” (lines 395-397). Please, the authors should homogenize the significant figures.

8) CONCLUSIONS. This section perfectly remarks the most relevant outcomes found by the authors in this work. The authors should add a brief statement to discuss about the future line actions to pursue this research and the open perspectives.

Comments on the Quality of English Language

The manuscript is generally well-written albeit it may be desirable if the authors could recheck it in order to polish those final details susceptible to be improved.

Author Response

Dear Editor and Reviewers,

We have read and carefully considered all the comments. We want to thank the Reviewers for their constructive suggestions and remarks and hope that our responses meet their expectations.

In the corrected version, we have highlighted (with' tracked changes') new sentences, rewritten parts, and made some other changes. Some of these changes were made because of another reviewer's comments.

Below, we provide our point-by-point response to the Reviewer's comments.

#Reviewer 1:

The manuscript titled “Pathological changes and metabolic adaptation in the myocardium of rats in response to chronic variable mild stress” by Ostrowska-Lésko, M.; et al. is a scientific work where the authors monitored the impact of mild stress factors on the myocardial metabolism and how these changes can eventually lead to heart diseases. The most relevant outcomes found by the authors could open new gates in the design of the next-generation of smart therapies to fight against heart malignancies. The manuscript is generally well-written and this is a topic of growing interest.

However, it exists some points that need to be addressed (please, see them below detailed point-by-point) to improve the scientifc quality of the submitted manuscript paper before this article will be consider for its publication in the International Journal of Molecular Sciences.

  1. I) KEYWORDS. The authors should consider to add the term “oxidative stress” in the keyword list.

 Thank you for the suggestion. ‘oxidative stress’ term was added to keywords.

2) INTRODUCTION. “Chronic variable mild stress (CVS) (…) depressive-like behavior (…) progression of cardiovascular diseases” (lines 30-40). Could the authors provide some quantitative insights according to the worldwide burdens of cardiovascular diseases? This will greatly aid the potential reader to better understand the significance to conduct this research.

We agree with the reviewer's opinion. The following paragraph was added:

„Cardiovascular diseases are the leading cause of death worldwide, with approxi-mately 17.9 million deaths per year, which is approximately 32% of all global deaths [4]. Among these fatalities, 85% are attributed to heart attacks and strokes. Chronic stress is a significant risk factor for cardiovascular disease. Research indicates that elevated levels of stress significantly raise the risk of heart disease by 27% [5]. People who have depression have a 64% higher chance of developing coronary artery disease [6].”

3) RESULTS. “2.2. Body weight assessment” (lines 86-95). Why did the authors only monitor the first 40 days after chronic variable mild stress exposure and not longer time scales? (Figure 9 in line 349 indicates that the rats are forced to die after these 40 days). A brief statement should be furnished in this regard.

We thank the Reviewer for this comment. The following statement was added to 4.2. section:

„The weight of the rats was measured over 40 days. The measurement time was limited by the planned sacrifice of the animals after the experiment.”

4) “2.5. Oxidative DNA damage” (lines 133-143). What is the detection limit of the methodology followed by the authors to detect the number of DNA abasic sites? Some information should be provided in this regard.

The detection limit of the number of abasic sites is 0-40 per 1x10^5 bp. The information was added to 4.6. section.

5) Figure 6 (line 166). The lateral scale bar should be added to the histological section images. Same comment for the Fig. 7 (line 174), and Fig. 8 (line 181).

The scale bars were added to Figures 6-8.

6) DISCUSSION. This section is well-structured. “NADPH plays a dual role in redox equilibrium (…) increase NADPH levels activate NADPH oxidase  (…) which causes endothelial dysfunction by raising superoxide anion levels (…) Each organism adapts at the molecular and biochemical levels to oxidative stress” (lines 257-268). I agree with the authors according the information stated in these statements albeit it may be also desirable to indicate how the detection of molecular levels of NADP+/NADPH [1] is crucial to better understand the inner cellular homeostasis with NADPH Oxidase enzyme which can lead to cardiovascular malignancies [2].

[1] Pérez-Domínguez, S.; et al. Nanomechanical Study of Enzyme: Coenzyme Complexes: Bipartite Sites in Plastidic Ferredoxin-NADP+ Reductase for the Interaction with NADP. Antioxidants 202211, 537. https://doi.org/10.3390/antiox11030537.

[2] Poznyak, A.V.; et al. NADPH Oxidases and Their Role in Atherosclerosis. Biomedicines 20208, 206. https://doi.org/10.3390/biomedicines8070206.

Imbalances in NADPH levels can directly affect the activity of NOX enzymes and the formation of reactive oxygen species (ROS). ROS generated from NOX may cause the uncoupling of endothelial nitric oxide synthase (eNOS) and impair mitochondrial function, resulting in persistent oxidative stress and the development of cardiovascular disorders such as hypertension, atherosclerosis, and heart failure.

The proposed source [2] has been added to the discussion:

„By monitoring NADP+/NADPH levels and dynamics, researchers can gain a deeper understanding of the complex connection between NADPH homeostasis, NOX enzyme activity, and the oxidative stress associated with cardiovascular illnesses. It is essential for the development of precise treatments that involve modulating NADP+/NADPH metabolism and redox balance.”

7) MATERIALS & METHODS. The authors perfectly explained all the steps “The supernatant (…) (pH 2.6; 3.1; 3.35; 4.05 and 4.65)” (lines 395-397). Please, the authors should homogenize the significant figures.

The author are unsure whether they completely comprehend the reviewer's remark. The pH entry has been corrected up to the second decimal place as follows: (pH 2.60; 3.10; 3.35; 4.05 and 4.65)

8) CONCLUSIONS. This section perfectly remarks the most relevant outcomes found by the authors in this work. The authors should add a brief statement to discuss about the future line actions to pursue this research and the open perspectives.

The following paragraph was added to 5. Conclusions section:

„Future research should focus on further elucidate the molecular pathways linking chronic stress to myocardial reprogramming. Key areas include detailed studies on the redirection of glycolysis, the activation of the pentose phosphate pathway, and the disturbances in myocardial amino acid metabolism. Exploring the potential of targeting FA oxidation and promoting glucose oxidation as therapeutic strategies for heart disease in the context of chronic stress may be viable. Developing and testing specific inhibitors of FA oxidation or agents that enhance glucose oxidation could lead to new treatments. Open perspectives involve understanding the broader implications of these findings for human health, particularly the common mechanisms underlying depression and cardiovascular diseases. This knowledge could contribute to innovative preventive and therapeutic approaches for individuals exposed to chronic stress, ultimately improving mental and cardiovascular health outcomes. Collaborative studies integrating clinical and preclinical research will be essential to translate these findings into effective interventions.”

We thank the Editor and the Reviewers for their time and effort in reviewing our manuscript.

Sincerely,

The Authors

Reviewer 2 Report

Comments and Suggestions for Authors

This manuscript describes that the pathological and function changes of heart in the chronic variable mild stress (CVS) rats, such as myocardial metabolism alterations, myocardial amino acid metabolism disturbances, glucose transport and metabolism gene expression changes. Overall, the finding in this manuscript is interesting, and would be helpful to the relevant research field. This reviewer recommends the acceptance for publication in Nutrients after the following minor issues addressed.

1.     Figures 1-4, the data should be also present as the scatter plot graph, to give more details.

2.     This manuscript should give the reason why the free amino acid concentration in heart was measured. The author should explain why only the four amino acids changes. 

3.     Figures 6-8, the scale bar should be included.

4.     In table 2, the p value cannot presented as p<0.000

Author Response

Dear Editor and Reviewers,

We have read and carefully considered all the comments. We want to thank the Reviewers for their constructive suggestions and remarks and hope that our responses meet their expectations.

In the corrected version, we have highlighted (with' tracked changes') new sentences, rewritten parts, and made some other changes. Some of these changes were made because of another reviewer's comments.

Below, we provide our point-by-point response to the Reviewer's comments.

  1. Figures 1-4, the data should be also present as the scatter plot graph, to give more details.

We thank the Reviewer for this comment. Before making the figures, the authors considered different types of graphs. According to the authors, scatter plot graph will make the results much less legible. The SD range reflects well the spread of results.

  1. This manuscript should give the reason why the free amino acid concentration in heart was measured. The author should explain why only the four amino acids changes. 

The following chapter was added to the discussion section:

Heart dysfunction is characterized by mitochondrial malfunction and a shift from using FAs to glucose for energy production. The heart’s response to chronic stress involves changes in amino acid metabolism, especially glucogenic amino acids that can be converted into intermediates of the citric acid cycle, leading to the production of glucose through gluconeogenesis. Moreover, certain amino acids may accumulate in heart failure, ischemic heart disease, or cardiomyopathies [31]. Here, we measured free amino acids concentration to provide insights into the heart’s response. In the chronic unpredictable mild stress (CUMS) model, the amino acids were reported to be altered in the heart tissue [32]. While the changes in concentration levels of amino acids observed in this study were not discussed concerning the heart, some general observations can be made. An increase in concentration of almost all glucogenic amino acids indicates a shift in metabolism towards producing glucose, either to meet energy demands or pathological factors as a response to CVS.

  1. Figures 6-8, the scale bar should be included.

Scale bars were added.

  1. In table 2, the pvalue cannot presented as p = 0.000

We thank the Reviewer for their thoroughness. P-value has been corrected as p <0.001.

We thank the Editor and the Reviewers for their time and effort in reviewing our manuscript.

Sincerely,

The Authors